# MITIGATING DIALOGUE HALLUCINATION FOR LARGE VISION LANGUAGE MODELS VIA ADVERSARIAL INSTRUCTION TUNING

## ABSTRACT

Mitigating hallucinations of Large Vision Language Models (LVLMs) is crucial to enhance their reliability for general-purpose assistants. This paper shows that such hallucinations of LVLMs can be significantly exacerbated by preceding user-system dialogues. To precisely measure this, we first present an evaluation benchmark by extending popular multi-modal benchmark datasets with prepended hallucinatory dialogues powered by our novel Adversarial Question Generator (AQG), which can automatically generate image-related yet adversarial dialogues by adopting adversarial attacks on LVLMs. On our benchmark, the zero-shot performance of state-of-the-art LVLMs drops significantly for both the VQA and Captioning tasks. Next, we further reveal this hallucination is mainly due to the prediction bias toward preceding dialogues rather than visual content. To reduce this bias, we propose Adversarial Instruction Tuning (AIT) that robustly fine-tunes LVLMs against hallucinatory dialogues. Extensive experiments show our proposed approach successfully reduces dialogue hallucination while maintaining performance.

## 1 INTRODUCTION

Developing a general-purpose assistant that interacts with humans through channels such as vision and language is one of the important problems in artificial intelligence. Inspired by the remarkable success of Large Language Models (LLMs), such as ChatGPT (Ouyang et al., 2022), the community has paid growing interest in developing *multi-modal* assistants, so-called Large Vision Language Models (LVLMs), that align vision foundation models (Chen et al., 2023; Radford et al., 2021) with LLMs to support visual-language instructions. Many LVLMs including LLaVA (Liu et al., 2023c), MiniGPT-4 (Zhu et al., 2023), and InstructBLIP (Dai et al., 2023) have shown powerful zero-shot generalization ability in various vision-language tasks such as classification (Pham et al., 2021; Park et al., 2024), detection (Li et al., 2022), visual question answering (VQA) (Song et al., 2022), and Captioning (Xu et al., 2023).

Despite their great success, several studies have revealed that LVLMs are prone to hallucination issues (Ji et al., 2023; Bang et al., 2023). While most studies focus on *object* hallucinations (Li et al., 2023b; Liu et al., 2023a), where LVLMs often answer inconsistently with contents of objects in a given image, the effect of user-system *dialogues* on hallucination has received little attention. Surprisingly, we found that such hallucinations can be significantly exacerbated by preceding user-system dialogues. For example, as shown in Figure 1(a), certain contents in preceding dialogues ("*eco-friendly*") conflicting with the current question can distract LVLMs, resulting in incorrect answers ("*Wood*"). This problem, which we call *dialogue hallucination*, is crucial in practice because a user usually interacts with the system through multi-round chats so that the user can unintentionally attack LVLMs in early chats and get unfaithful answers in later chats.

In this paper, we first present an evaluation benchmark, EvalDial, to more precisely measure the dialogue hallucination of LVLMs. Our benchmark is constructed on popular vision-language benchmark datasets for VQA and Captioning tasks (Xu et al., 2023). Specifically, for each *test* example in each dataset, we create corresponding hallucinatory dialogues that can be prepended to the original test question. Moreover, to mimic actual user behaviors interacting with the assistant within visual contexts of a given image, we further introduce Adversarial Question Generator (AQG), which automatically generates *image-related* yet *adversarial* dialogues, by steadily incorporating an extra LVLM into the black-box optimization of adversarial attack (Ilyas et al., 2018; Maus et al., 2023). With

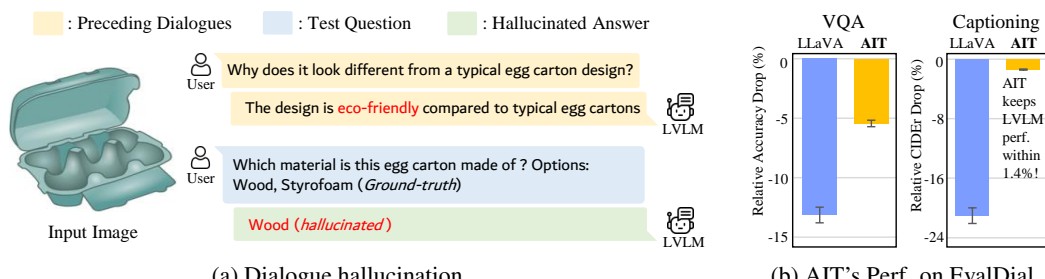

(a) Dialogue hallucination.

(b) AIT's Perf. on EvalDial.

Figure 1: (a) shows an example of dialogue hallucination generated by an LVLM (e.g., LLaVA (Liu et al., 2023c)) for a test example in ScienceQA dataset; (b) shows the average performance drop of LLaVA and AIT on EvalDial for VQA and Captioning tasks with prepended adversarial dialogues.

optimization, AQG can generate effective adversarial questions, while GPT-4 or other textual-based red teaming methods struggle to generate such subtle cases. On EvalDial, the zero-shot performance of state-of-the-art LVLMs drops by up to 37.7% for the VQA task and 59.6% for the Captioning task.

To mitigate the dialogue hallucination, we conduct input token attention analysis and embedding distribution analysis. We find that the hallucination is mainly due to the prediction *bias* to preceding dialogues rather than visual contents. Therefore, we propose Adversarial Instruction Tuning (AIT) that aims to reduce such prediction bias by robustly fine-tuning LVLMs on augmented visual-instruction datasets with hallucinatory dialogues. Specifically, we introduce masked instruction tuning to focus on the target answers instead of hallucinatory responses from adversarial dialogues. Extensive experiments on six vision-language datasets in EvalDial demonstrate that AIT successfully reduces the dialogue hallucination while maintaining the performance of LVLM for both VQA and Captioning tasks, as shown in Figure 1(b).

Our main contributions can be summarized as:

- We find that LVLMs are prone to hallucination by preceding dialogues.
- We present an evaluation benchmark (EvalDial) for dialogue hallucination with a novel adversarial question generator (AQG).
- We reveal LVLM's prediction bias toward hallucinatory dialogues by input token attention analysis.
- We propose AIT with masked instruction tuning that successfully reduces the dialogue hallucination on many vision-language datasets.

## 2 RELATED WORK

### 2.1 INSTRUCTION-FOLLOWING LVLMs

Instruction-tuning LLMs such as GPT (Brown et al., 2020) have significantly enhanced their zero-shot generalization ability in various NLP tasks (Wang et al., 2022), resulting in instruction-following LLMs such as ChatGPT (Ouyang et al., 2022). Recently, this instruction-tuning idea has been actively extended to *vision-language* domains, and many instruction-following LVLMs have been developed (Bai et al., 2023; Liu et al., 2023c; Zhu et al., 2023; Dai et al., 2023; Wang et al., 2023; Achiam et al., 2023; Team et al., 2023). In general, most LVLMs combine pre-trained vision encoders (e.g., CLIP (Radford et al., 2021)) with LLMs by fine-tuning them on visual-language instruction datasets (Zhang et al., 2023; Koh et al., 2023). Notably, LLaVA (Liu et al., 2023c) projects CLIP to LLaMA (Touvron et al., 2023), and then fine-tunes the models with a projection layer on a visual instruction dataset (Peng et al., 2023). Similarly, MiniGPT-4 (Zhu et al., 2023) uses BLIP-2 (Li et al., 2023a) as visual encoder and Vicuna as language decoder, and InstructBLIP (Dai et al., 2023) uses Q-former as the projection layer. These models have shown a powerful zero-shot performance in various vision-language tasks including VQA and Image Captioning (Xu et al., 2023).

### 2.2 HALLUCINATIONS OF LVLMs

LVLMs are prone to hallucination issues as their output descriptions are often inconsistent with the input images and text instructions (Ji et al., 2023; Bang et al., 2023; Bai et al., 2024). Most prior work focuses on *object* hallucination where the output descriptions of objects are non-existent or inaccurate from the given image (Rohrbach et al., 2018; Zhou et al., 2023). Many evaluation benchmarks for object hallucination have been proposed (Gunjal et al., 2024; Sun et al., 2023). POPE (Li et al., 2023b)

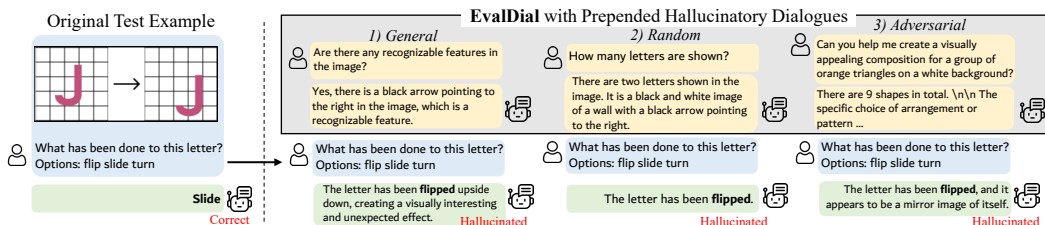

Figure 2: Overview of dialogue hallucinations on **EvalDial**. A test example on IconQA that LLaVA originally answers correctly becomes hallucinated after three types of prepended dialogues, i.e., General, Random, and Adversarial.

converts the hallucination detection as a binary classification, GAVIE (Liu et al., 2023a) leverages GPT-4 to evaluate the hallucination, and THRONE (Kaul et al., 2024) addresses hallucinations in open-ended free-form generations. To mitigate this, many works tried to enrich the visual-instruction datasets. LRV-Instruction (Liu et al., 2023a) reveals existing visual-instruction datasets are biased to positive responses, so they append instructions with negative responses in robust fine-tuning. HalluciDocter (Yu et al., 2023) introduces a hallucination cross-checking paradigm that can recover visual-instruction data. HACL (Jiang et al., 2023) proposes a hallucination-augmented contrastive learning framework. Note that, while some recent literature investigates the effect of deceptive prompts and dialogue hallucination on LVLMs (Shi et al., 2023; Qian et al., 2024; Chen et al., 2024; Cao et al., 2024), they only provide *hand-crafted* or *singular domain* evaluation datasets.

## 2.3 ADVERSARIAL ATTACKS ON LANGUAGE MODELS

Adversarial attacks aim to ruin output predictions of a model by perturbing the input examples (Chakraborty et al., 2018; Ilyas et al., 2018). For attacking LLMs, AdvPrompt (Maus et al., 2023) finds adversarial prompts to generate nonsensical text by increasing the perplexity of the output tokens. GCG (Zou et al., 2023) obtains adversarial suffix prompts to generate objectionable behavior, such as harmful content. Harmbench (Mantas et al., 2024) proposed a framework for large-scale automated red teaming methods and defence of LLMs. However, since such generated adversarial prompts are *incomprehensive* to humans, e.g., a sequence of random letters, they are *not* directly applicable to generating adversarial dialogues for LVLMs that must be in natural language.

## 3 DIALOGUE HALLUCINATION AND EVALUATION BENCHMARK

We first formulate an LVLM and its dialogue hallucination. Then, we describe EvalDial, a benchmark we release to evaluate dialogue hallucination, powered by our novel Adversarial Question Generator.

## 3.1 DIALOGUE HALLUCINATION OF LVLMs

**Instruction-following LVLM.** For an input image $X_v$ and a user question $X_q$, an *instruction-following LVLM* $f_{LVLM}$ aims to generate a corresponding output text answer $X_a$. For multi-round conversation, the $t$-th round answer $X_a^t$ can be formulated as:

$$X_a^t = f_{LVLM}(X_v, X_{dialogue}^{<t}, X_q^t),\qquad(1)$$

where $X_{dialogue}^{<t} = (X_q^1, X_a^1, \cdots, X_q^{t-1}, X_a^{t-1})$ is a sequence of all previous dialogues before asking the $t$-th round question $X_q^t$. Here, we denote $i$-th round dialogue $X_{dialogue}^i = (X_q^i, X_a^i)$ as a pair of a user question and the corresponding answer from LVLM at round $i$.

**Dialogue Hallucination.** According to hallucination literature (Ji et al., 2023; Bang et al., 2023), the most inclusive and standard definition of hallucinations is "the generated content that is nonsensical or unfaithful to the given source content". Based on this, we define the *dialogue hallucination* of LVLMs as in Definition 3.1.

**Definition 3.1.** (DIALOGUE HALLUCINATION) We call a generated answer $\tilde{X}_a$, that is faithful *without* any dialogue but becomes unfaithful *after* some preceding dialogues, *dialogue hallucination*. That is, the output answer $\tilde{X}_a$ hallucinated by the preceding dialogues is represented as $f_{LVLM}(X_v, X_{dialogue}^{<t}, X_q^t) = \tilde{X}_a$ while $f_{LVLM}(X_v, X_q^t) = X_a$, where $X_a$ represents the originally non-hallucinated answer. $\square$

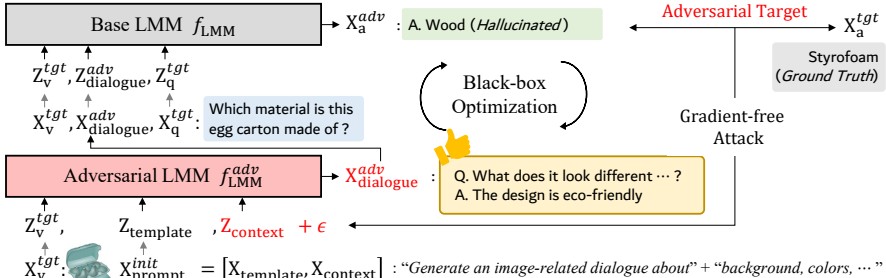

Figure 3: shows the overview of AQG, generating an adversarial dialogue $X_{\texttt{dialogue}}^{adv}$ (in yellow box) to hallucinate the answer $X_a^{adv}$ (in green box) by incorporating an extra LVLM into the optimization of adversarial attack

Note that, dialogue hallucination can include various types of generated contents, such as wrong answers for VQA (Ji et al., 2023), inaccurate descriptions for Captioning (Xu et al., 2023), and responses of non-existent contents for Object-finding (Li et al., 2023b).

## 3.2 EVALDIAL: AN EVALUATION BENCHMARK

We construct EvalDial on top of popular vision-language *test* datasets; ScienceQA (Lu et al., 2022), OKVQA (Marino et al., 2019), GQA (Hudson and Manning, 2019), and IconQA (Lu et al., 2021) datasets for VQA task, and NoCaps (Agrawal et al., 2019), Flickr-30k (Plummer et al., 2015), and WHOOPS (Bitton-Guetta et al., 2023) datasets for Captioning task. For each test example in each dataset, we create three types of dialogue, i.e., *General*, *Random*, and *Adversarial*, that are prepended into the original test question or instruction. Figure 2 illustrates more details of EvalDial.

1. **General Dialogue** contains a general question, that can be universally asked to any image, and its corresponding answer is obtained from LVLM. For example, a general dialogue can be *"Q. What is the dominant color in the image? A. It's blue"*. We extract 10 general questions from GPT by prompting *"Generate 10 general questions for a random image"*. See Appendix A for details.

2. **Random Dialogue** consists of a pair of random questions, that are completely irrelevant to a given image, and its corresponding answer obtained from LVLM. For example, given a car image, a random dialogue can be *"Q. what kind of animals are these? A. there are no animals"*. To generate such questions, we randomly extract questions from the VQA-v2 dataset (Goyal et al., 2017), which does not have an overlapping set of questions with the aforementioned benchmark test datasets.

3. **Adversarial Dialogue** contains an *image-related* yet *adversarial* question that causes hallucinations to the original test question. Because real users often have chats related to the context of the given image, it is essential to verify LVLM's robustness against the image-related but adversarial dialogue. However, generating such subtle questions is very challenging. Thus, we propose AQG, an adversarial question generator based on black-box adversarial attack techniques (Andriushchenko et al., 2020; Maus et al., 2023) as elaborated in Section 3.3. Detailed generated adversarial questions are in Appendix B.

Note that, for all three types of dialogues, EvalDial only contains questions without corresponding answers, since the answers are naturally generated by LVLMs in the test phase. Evaluation results of state-of-the-art LVLMs on EvalDial can be found in Section 5.

## 3.3 ADVERSARIAL QUESTION GENERATOR

To mimic real-world user-system interactions, the adversarial dialogues should be *image-related* and *natural-sounding*, yet *adversarial*. However, automatically generating such subtle dialogues in any context is very challenging, because LVLMs usually do not know when they hallucinate, which means it is difficult to obtain these adversarial dialogues by simply prompting (Gunjal et al., 2024). Therefore, as in Figure 3, we propose AQG that can automatically generate natural-sounding adversarial questions by adopting adversarial attack techniques with an extra LVLM. Overall, AQG consists of two common components in adversarial attack; (1) threat model and (2) adversarial target.

**Threat Model.** A threat model represents a specific type of attack, *e.g.*, $l_2$-bounded noise for image classification (Andriushchenko et al., 2020), or token-restricted prompt for language models (Maus

---

**Algorithm 1** Adversarial Question Generator (AQG)

---

INPUT: $X_{\text{prompt}}^{init}$: initial prompt, $X_{\text{dialogue}}^{adv}$: generated adversarial dialogue, $X_{\text{a}}^{adv}$: output answer, and $X_{\text{a}}^{tgt}$: target answer
1: Initialize $X_{\text{prompt}} \leftarrow [X_{\text{template}}); X_{\text{context}})]$; $\ell^{tgt} \leftarrow 0$; $\sigma \leftarrow 0.1 * \text{AvgDist}$
2: **for** $i = 1$ **to** $r$ **do**
3:     $\epsilon \sim \mathcal{N}(0, \sigma)$
4:     $X_{\text{dialogue}}^{adv} = f_{\text{LVLM}}^{adv}(X_{\text{v}}^{tgt}, Z_{\text{template}}, Z_{\text{context}} + \epsilon)$    /* Dialogue Generation */
5:     $X_{\text{a}}^{adv} = f_{\text{LVLM}}(X_{\text{v}}^{tgt}, X_{\text{dialogue}}^{adv}, X_{\text{q}}^{tgt})$      /* Answer Generation */
6:     **if** $\mathcal{L}(X_{\text{a}}^{adv}, X_{\text{a}}^{tgt}) > \ell^{tgt}$ **do**
7:        $Z \leftarrow Z + \epsilon, \ell^{tgt} = \mathcal{L}(X_{\text{a}}^{adv}; X_{\text{a}}^{tgt})$     /* Updating Token Embedding with Gaussian Noise */
OUTPUT: Final adversarial dialogue $X_{\text{dialogue}}^{adv}$

---

et al., 2023). Then, the threat model of AQG should be confined to image-related and natural-sounding questions. To meet this requirement, AQG leverage an extra LVLM $f_{\text{LVLM}}^{adv}$ and force it to generate image-related and natural-sounding dialogues by only updating its prompt token embeddings $Z_{\text{prompt}} = \text{tokenize}(X_{\text{prompt}})$, where $X_{\text{prompt}}$ is an input prompt of $f_{\text{LVLM}}^{adv}$.

The adversarial prompt $X_{\text{prompt}}$ consists of a *fixed* template prompt $X_{\text{template}}$, *e.g.*, "*generate an image-related dialogue about*", concatenated with an *updatable* context prompt $X_{\text{context}}$ initialized as "*background, colors, history, etc*", such that $X_{\text{prompt}}^{init} = [X_{\text{template}}; X_{\text{context}}]$.

In optimization, we only perturb the context prompt by injecting a random noise $\epsilon$ into the context token embeddings $Z_{\text{context}}$. The random noise $\epsilon$ is sampled from a Gaussian distribution with the mean of 0 and the standard deviation $\sigma = 0.1 * \text{AvgDist}$, where AvgDist is the average distance between embeddings of all possible tokens, which is shown to be effective in attacking language models (Maus et al., 2023).

**Adversarial Target.** We use the *negative* sentence similarity between the target answer $X^{tgt}$ and generated answer $X_{\text{a}}^{adv}$ as the adversarial target. Formally, our adversarial target can be denoted as $\mathcal{L} = -\text{Sim}(X^{tgt}, X_{\text{a}}^{adv})$, where we use CIDEr (Vedantam et al., 2015) score as the similarity function.

**Optimization Procedure.** Algorithm 1 details the overall optimization process of AQG, which is self-explanatory. AQG finds the best adversarial dialogue $X_{\text{dialogue}}^{adv}$ that maximizes target loss $\ell^{tgt}$ by iteratively updating better random noise $\epsilon$. See Appendix C for a more detailed description.

## 4 ADVERSARIAL INSTRUCTION TUNING

We first provide an input token attention analysis to help understand dialogue hallucination. Based on this, we present a more robust instruction tuning paradigm, *Adversarial Instruction Tuning (AIT)*.

### 4.1 INPUT TOKEN ATTENTION ANALYSIS

Input feature attention analysis is a popular method to investigate the contribution of input features to model prediction, *e.g.*, GradCAM (Selvaraju et al., 2017) for vision models, or token attention map (Sundararajan et al., 2017; Kokhlikyan et al., 2020) for language models. Here, we introduce a new attention-based metric for LVLM, coined Dialogue Tokens Attention Ratio (DTAR), that helps to analyze the dialogue hallucination in instruction-following LVLMs, by calculating the contribution of preceding dialogues to output answer prediction.

**Dialogue Tokens Attention Ratio.** Let $Z_{\text{v}} = W_{\text{Proj}} \cdot f_{\text{VE}}(X_{\text{v}})$ be token embeddings of input image $X_{\text{v}}$, where $W_{\text{Proj}}$ is a linear projection layer that convert the output patches of visual encoder $f_{\text{VE}}$ to input tokens of LLM, and $Z_{\text{dialogue}} = f_{\text{token}}(X_{\text{dialogue}})$ be token embeddings of input preceding dialogue $X_{\text{dialogue}}$. Also, let $P(X_{\text{a}})$ be the probability of output answer tokens. Then, we define *Dialogue Tokens Attention Ratio (DTAR)* using the *gradient* of input token embeddings $Z_{\text{v}}$ and $Z_{\text{dialogue}}$ with respect to the output token probability $P(X_{\text{a}})$, as in Definition 4.1.

**Definition 4.1.** (DIALOGUE TOKENS ATTENTION RATIO) For each instruction example, DTAR is the ratio of the sum of all absolute attention scores of dialogue tokens over that of all input tokens;

$$\sum_i \left[ \sum_j \left| \frac{\partial P(X_{\text{a},i})}{\partial Z_{\text{dialogue},j}} \right| / \left( \sum_j \left| \frac{\partial P(X_{\text{a},i})}{\partial Z_{\text{dialogue},j}} \right| + \sum_k \left| \frac{\partial P(X_{\text{a},i})}{\partial Z_{\text{v},k}} \right| \right) \right], \tag{2}$$

Table 1: Summary of DTAR scores for correct (non-hallucinated) and hallucinated cases.

| Models | Cases | Mean | Std |
|--------|-------|------|-----|
| LLaVA | Non-hallucinated | 0.19 | 0.06 |
| | Hallucinated | 0.37 | 0.11 |
| AIT | Non-hallucinated | 0.17 | 0.09 |
| | Hallucinated | 0.25 | 0.12 |

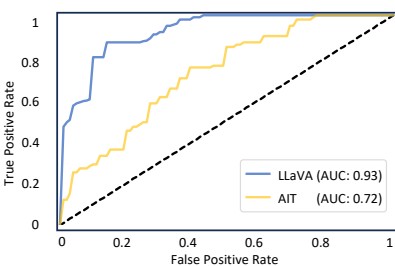

Figure 4: AUC-ROC comparison.

where $X_{\mathrm{a},i}$ denotes $i$-th token in the output answer $X_\mathrm{a}$, $Z_{\mathrm{dialogue},j}$ denotes $j$-th token embedding in $Z_{\mathrm{dialogue}}$, and $Z_{\mathrm{v},k}$ denotes $k$-th token embedding in $Z_\mathrm{v}$. Intuitively, DTAR means the contribution of preceding dialogues over the input image to output the final answer. $\qquad\square$

**DTAR Analysis for Dialogue Hallucination.** Using LLaVA (Liu et al., 2023c), we calculate the DTAR score of hallucinated examples and that of non-hallucinated examples on EvalDial built on ScienceQA dataset. We select 500 hallucinated examples by preceding adversarial dialogues and 500 non-hallucinated examples, then calculate the DTAR score for each example. Table 1 summarizes the mean and standard deviation of DTAR scores for non-hallucinated and hallucinated cases. For LLaVA, the DTAR score of hallucinated examples is higher than that of non-hallucinated examples, meaning that LLaVA focuses more on preceding dialogues than image features for the prediction of the hallucinated case. Similarly, Figure 4 shows the AUC-ROC curves of DTAR score on two cases, hallucinated and non-hallucinated. The AUC of DTAR score of LLaVA is 0.935, which is high, meaning that LLaVA often relies on hallucinatory dialogue for prediction, thereby causing hallucinations. Section 4.2 is proposed to suppress this prediction bias toward hallucinatory dialogues.

## 4.2 ADVERSARIAL INSTRUCTION TUNING (AIT)

To reduce the negative effect of hallucinatory dialogues, we propose AIT to perform instruction tuning on adversarially augmented visual instruction datasets. AIT first generates hallucinatory dialogues and injects them into visual instruction training datasets, and then performs instruction tuning by masking the hallucinatory dialogues in loss calculation.

**Hallucinatory Dialogue Generation.** We create hallucinatory dialogues following the protocol of EvalDial in Section 3.2. Specifically, for each training example of the visual-instruction dataset such as LLaVA-Instruct-665k (Liu et al., 2023b), we generate hallucinatory questions $X_\mathrm{q}^{i,adv}$ in order to hallucinate each round's question $X_\mathrm{q}^i$ in the original training example $X_{\mathrm{dialogue}}^{<t}$, and generate its corresponding answers $X_\mathrm{a}^{i,adv}$ by simply asking the hallucinatory questions to LVLMs. The hallucinatory question includes all types of dialogues, i.e., General, Random, and Adversarial. For the training examples with $t$ instruction rounds, we randomly augment $m$ rounds out of $t$ rounds.

**Hallucinatory Dialogue Injection.** For each training example, the $i$-th round dialogue $X_{\mathrm{dialogue}}^i = (X_\mathrm{q}^i, X_\mathrm{a}^i)$ can be adversarially augmented by prepending a hallucinatory dialogue $X_{\mathrm{dialogue}}^{i,adv} = (X_\mathrm{q}^{i,adv}, X_\mathrm{a}^{i,adv})$ as follows:

$$X_{\mathrm{aug}}^i = (X_{\mathrm{dialogue}}^{i,adv}, X_{\mathrm{dialogue}}^i). \qquad (3)$$

That is, if $m = 1$ and $i$-th round instruction is chosen to be augmented, then the overall augmented input $X_{\mathrm{aug}}$ for LVLM are formulated as,

$$X_{\mathrm{aug}} = (X_\mathrm{v}, X_{\mathrm{dialogue}}^1, \cdots, X_{\mathrm{aug}}^i, \cdots, X_{\mathrm{dialogue}}^t). \qquad (4)$$

**Masked Instruction Tuning.** As opposed to standard instruction tuning, where LVLM minimizes the cross-entropy loss of answer tokens in all rounds of dialogues, we *mask* answer tokens of hallucinatory dialogues so that they are not factored into calculating the cross-entropy loss. Therefore, the LVLM is not trained to generate answers in hallucinatory dialogues and can be robustly trained to generate correct answers to subsequent questions despite the presence of preceding hallucinatory dialogues.

Table 2: Zero-shot performance of LVLMs on EvalDial with prepended three types of single-round dialogues, General (Gen), Random (Rand), and Adversarial (Adv). We compare AIT with LLaVA and highlight better performance against dialogue hallucinations in bold. The average relative performance drop (% Avg Drop) from the None-dialogue case for each LVLM is also presented.

| Datasets | MiniGPT-4 (7B) | | | | InstructBLIP (7B) | | | | LLaVA-v1.5 (7B) | | | | AIT (7B) | | | |
|---|---|---|---|---|---|---|---|---|---|---|---|---|---|---|---|---|
| | None | Gen | Rand | Adv | None | Gen | Rand | Adv | None | Gen | Rand | Adv | None | Gen | Rand | Adv |
| OKVQA | 36.4 | 28.4 | 24.7 | 24.2 | 60.0 | 57.4 | 59.2 | 53.7 | 54.8 | 54.4 | 53.6 | 48.4 | 56.8 | **59.6** | **55.2** | **53.0** |
| GQA | 31.2 | 26.2 | 19.4 | 18.8 | 50.4 | 49.0 | 46.8 | 46.2 | 55.8 | 55.4 | **57.0** | 49.0 | 57.8 | **56.0** | 55.4 | **55.6** |
| IconQA | 37.2 | 31.0 | 24.0 | 22.4 | 53.0 | 52.2 | 51.6 | 51.1 | 48.8 | **45.8** | 46.4 | 41.2 | 47.8 | 45.4 | **49.2** | **45.0** |
| % Avg Drop | - | −18.2 | −35.1 | −37.7 | - | −2.9 | −3.7 | −7.5 | - | −2.5 | −1.7 | −13.1 | - | **−1.0** | **−1.3** | **−5.4** |
| NoCaps | 40.0 | 34.4 | 31.9 | 21.5 | 45.7 | 26.7 | 27.5 | 21.8 | 42.1 | 41.2 | 40.8 | 35.8 | 53.3 | **53.0** | **52.6** | **52.9** |
| Flickr-30K | 27.2 | 23.9 | 18.4 | 16.9 | 49.3 | 22.4 | 23.0 | 19.5 | 31.0 | 30.4 | 29.6 | 19.9 | 39.5 | **38.8** | **38.2** | **38.7** |
| WHOOPS | 48.0 | 45.3 | 44.6 | 25.7 | 73.4 | 27.6 | 30.7 | 25.0 | 39.7 | 38.5 | 38.7 | 34.8 | 42.8 | **42.3** | **40.5** | **42.2** |
| % Avg Drop | - | −10.6 | −19.9 | −43.5 | - | −52.8 | −50.4 | −59.6 | - | −2.4 | −3.4 | −21.0 | - | **−0.8** | **−3.3** | **−1.4** |

## 5 EXPERIMENTS

**Datasets.** Followed by Section 3.2, we use our proposed benchmark, EvalDial, for evaluating dialogue hallucination. We mainly use EvalDial built on top of OKVQA (Marino et al., 2019), GQA (Hudson and Manning, 2019), and IconQA (Lu et al., 2021) for VQA task, and NoCaps (Agrawal et al., 2019), Flickr-30K (Plummer et al., 2015), and WHOOPS (Bitton-Guetta et al., 2023) for Captioning task.

**Algorithms.** We compare the zero-shot performance of AIT with three recently proposed LVLMs: (1) MiniGPT-4 (7B) (Zhu et al., 2023), (2) InstructBLIP (7B) (Dai et al., 2023), and (3) LLaVA-v1.5 (7B) (Liu et al., 2023b).

**Implementation Details.** AIT uses the same model architecture with LLaVA-v1.5. For the hyper-parameters of adversarial instruction tuning, we train AIT for 1 epoch with a batch size of 128, and an initial learning rate of 2e-5 with a cosine scheduler. For hallucinatory dialogue injection, we generate adversarial dialogue examples from LLaVA-Instruct-80k, OKVQA with 9K examples, GQA with 15K examples, IconQA with 29K examples, and 0.5K examples each from NoCaps and Flickr-30K, which are mostly originally included in LLaVA-Instruct-665K. All methods are implemented with PyTorch 1.8.0 and executed on multiple NVIDIA A100 GPUs. Generating an adversarial dialogue for each image-QA example using AQG takes approximately 50 seconds on a single A100 GPU. By applying quantization, this can be reduced to about 18 seconds. The code is available at `https://github.com/dongmean/LVLM_DialHalu`.

**Evaluation.** For VQA task, we use top-1 accuracy that validates whether the ground-truth answer is in the generated sentence. For Image Captioning task, we use CIDEr (Vedantam et al., 2015) score, a popular metric to evaluate image captioning quality (Xu et al., 2023).

### 5.1 MAIN RESULTS ON EVALDIAL

**Efficacy of AQG.** Table 2 summarizes the zero-shot performance of LVLMs on EvalDial. Overall, with three types of dialogues prepended, the performance of all existing baselines such as LLaVA, MiniGPT-4, and InstructBLIP are significantly degraded by up to 37.7% for VQA task and 59.6% for Captioning task. Among the three types of dialogues, adversarial dialogues generated by AQG show the highest performance drops for every baseline LVLM, which demonstrates the efficacy of AQG.

**Efficacy of AIT.** While every baseline LVLM is vulnerable to dialogue hallucinations, AIT maintains the most robust VQA and Captioning performance against dialogue hallucinations. Numerically, for VQA task with OKVQA, GQA, and IconQA datasets, AIT maintains VQA accuracy within −1.0% to −5.4% drops, while LLaVA drops by −2.5% to −13.1%. Similarly, for Captioning task with Nocaps, Flickr-30K, and WHOOPS datasets, AIT maintains Captioning performance within −0.8% to −1.4% drops, while LLaVA drops by −2.4% to −21.0%. Additionally, we evaluated using FAITHSCORE (Jing et al., 2023) on the Captioning task, and the results are shown in Appendix D.

### 5.2 IN-DEPTH ANALYSIS OF AQG

**Superiority of AQG over Possible Attacking Methods.** We compare the effectiveness of AQG in attacking subsequent target questions, i.e., triggering dialogue hallucinations. Since there is no

Table 3: Comparison of AQG with different attacking methods. The lower, the more effective in attacking.

| Dataset | LLaVa-v1.5 | | | | | | |
| --- | --- | --- | --- | --- | --- | --- | --- |
| | None | GPT4 | GCG | $GCG^m$ | PAIR | TAP | **AQG** |
| GQA | 55.8 | 54.8 | 54.4 | 55.4 | 57.0 | 62.4 | **49.0** |
| OKVQA | 54.8 | 54.4 | 53.4 | 53.6 | 55.4 | 64.0 | **48.4** |
| IconQA | 48.8 | 46.6 | 44.2 | 43.8 | 42.6 | 44.8 | **41.2** |

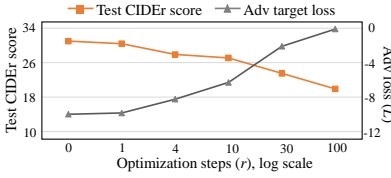

Figure 5: Effect of optimization steps in AQG to attack LLaVA on Flickr.

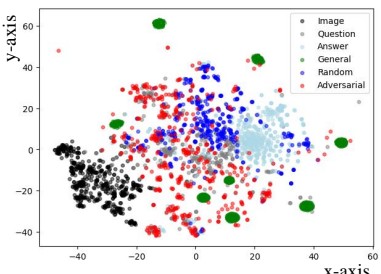

(a) TSNE plot of dialogue examples.

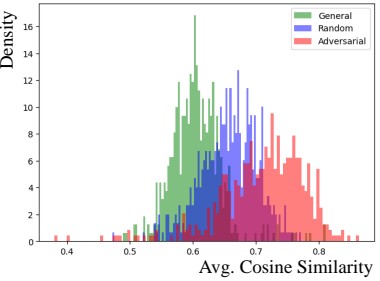

(b) Density plot of three dialogue types.

Figure 6: Embedding distribution analysis for three types of dialogues. (a) illustrates the TSNE plot of each dialogue example with its corresponding target image, question, and answer on the joint text-image embedding space of InstructBLIP. (b) shows the density plot of three types of dialogue in terms of the average cosine similarity to the corresponding target image, question, and answer.

comparative attacking approach fit to the dialogue hallucination problem for LVLM, we adopt *one* GPT-prompting based attacking approach, and *four* text-based attacking methods adopted from Harmbench (Mantas et al., 2024), a hallucination generation framework for LLM. For the GPT-prompting attack, we generate adversarial questions by carefully asking GPT to generate hallucinatory questions against the given target questions if prepended (See Appendix E for prompting details). For text-based attacking methods, we use 4 different attacking variants in Harmbench, such as GCG, GCG-Multi ($GCG^m$), PAIR, and TAP.

Table 3 shows the performance of AQG compared to other attacking methods. Overall, AQG is the most effective in attacking the original test answer in three VQA datasets, including GQA, OKVQA, and IconQA. In detail, although we carefully ask GPT-4 to generate adversarial questions that can cause hallucination in the subsequent target questions, the generated questions do not degrade the performance of LLaVA; the performance is similar to the None case, where no dialogues are prepended. Also, other text-based attacking methods, from GCG to TAP, are not effective in attacking and sometimes fail to induce the hallucinations. However, AQG is consistently effective in attacking the original questions, showing the necessity of our optimization-based attack by understanding multi-modal semantics.

**Effect of Optimization Steps in AQG.** Figure 5 shows the effect of optimization steps in AQG to attack LLaVA's captioning performance on Flickr dataset. With more optimization steps in Figure 5, AQG generates more effective adversarial dialogues with higher target loss, thereby successfully attacking the original test captioning performance. This indicates the adversarial objective and optimization process in is well-designed and appropriate to generate better adversarial dialogues in multi-modal semantics.

**Distributional Analysis of Adversarial Examples Generated by AQG.** Figure 6 shows the distributions of three types of dialogues, including the adversarial dialogues generated by AQG, with two types of plots: (1) TSNE plot; and (2) density plot. In the TSNE plot, we visualize all the embeddings of the target image, question, and answer with General, Random, and Adversarial dialogues on a joint embedding space of InstructBLIP. Overall, the General (green) dialogues tend to be located far from the target image, question, and answer (black, grey, and light blue, respectively), while the Random (blue) and Adversarial (red) dialogues appear to be located closer to the target image, question, and answer. For a more detailed sample-level analysis, in the density plot, we measure the average cosine similarity from each dialogue sample to its corresponding target image, question, and answer, and plot the density. The Adversarial dialogues have the highest average similarity with their

Table 4: Effect of applying masked instruction tuning during AIT on IconQA dataset.

| Model | IconQA | | | |
|---|---|---|---|---|
| | None | Gen | Rand | Adv |
| LLaVA | 48.8 | 45.8 | 46.4 | 41.2 |
| AIT (Unmasked) | 32.8 | 30.6 | 33.6 | 29.8 |
| AIT (Masked) | **47.2** | **49.0** | **48.0** | **47.8** |

Table 5: Effect of the number of hallucinatory dialogues ($m$) used during AIT. Each AIT model is augmented and finetuned from the LLaVA-Instuct-150K dataset.

| Model | GQA | | | | IconQA | | | |
|---|---|---|---|---|---|---|---|---|
| | None | Gen | Rand | Adv | None | Gen | Rand | Adv |
| $\text{AIT}_{m=1}$ | 44.2 | 38.8 | 38.8 | 35.6 | 55.0 | 46.4 | 47.2 | 40.4 |
| $\text{AIT}_{m=2}$ | 44.4 | 39.2 | 36.6 | 35.8 | 61.0 | 44.0 | 47.8 | 41.6 |
| $\text{AIT}_{m=all}$ | **45.6** | **39.6** | **38.9** | **36.8** | **68.6** | **47.6** | **51.4** | **50.0** |

Table 6: Effect of multi-round prepended dialogues on LVLMs using GQA dataset.

| Round | LLaVA-v1.5 | | | AIT | | |
|---|---|---|---|---|---|---|
| | Gen | Rand | Adv | Gen | Ran | Adv |
| 1 | 55.4 | 57.0 | 49.0 | 56.0 | 55.4 | 55.6 |
| 2 | 52.4 | 54.0 | 48.8 | 53.4 | 55.4 | 55.0 |
| 4 | 53.4 | 53.2 | 48.5 | 52.6 | 55.5 | 55.2 |

Table 7: Sanity check results; Effect of prompt length on the model performance.

| # Repeats (N) | LLaVA-v1.5 | | | |
|---|---|---|---|---|
| | 0 | 1 | 2 | 4 |
| OKVQA | 54.8 | 54.8 | 54.8 | 54.7 |
| GQA | 55.8 | 55.8 | 55.7 | 54.8 |
| IconQA | 48.8 | 48.8 | 48.8 | 48.7 |

corresponding images, questions, and answers, indicating dialogues that are semantically relevant to the Image, Question, and Answer together cause higher levels of hallucination.

## 5.3 ABLATION STUDIES OF AIT

**Effect of Loss Masking on Prepended Adversarial Dialogues.** Table 4 shows the effect of the loss masking on prepended adversarial dialogues in masked instruction tuning. Without loss masking, AIT shows unsatisfactory performance in mitigating dialogue hallucination. This is because the fine-tuned LVLM without loss masking is forced to generate the answer even for injected random or adversarial dialogues, which are out-of-context from the given image or harmful in maintaining the context of the given image, resulting in more hallucinations in later rounds of chats.

**Effect of The Number of Injected Hallucinatory Dialogues.** As elaborated in Section 4.2, AIT randomly chooses $m$ rounds for each training example in the visual-instruction dataset to inject hallucinatory dialogues. Here, we investigate the effect of the number $m$ of injected hallucinatory dialogues. We set $m$ to be 1,2 and all from available dialogues per example since most examples contain 4 rounds of dialogues on average. To control the effect of data size in the study, we only use LLaVA-Instruct-150K for fine-tuning. As in Table 5, with more injected adversarial examples used during AIT, the model gets more robust to the adversarial attack. Therefore, the more hallucinatory dialogues injected into adversarial instruction tuning, the more performance gain we can have.

More studies of the object hallucination baseline (Liu et al., 2023a) on EvalDial are in Appendix F.

## 5.4 RESULTS ON MULTI-ROUND EVALDIAL

As users usually interact with LVLMs via multi-round dialogues, it is essential to explore the effectiveness of LVLMs in multi-round cases. For General and Random type attacks, we randomly sample one question for each round and prepend them before the target question. For the Adversarial type attack, we generated different adversarial questions by using AQG based on the target question-answer pair for each round. Table 6 shows the results with 1,2, and 4 rounds of prepended dialogues during evaluation on the LLaVA baseline and AIT model. The results indicate that more prepended questions make the model less likely to answer the target question correctly. However, AIT shows a more robust performance than the baseline on different types of prepended dialogues. Specifically, as the round of prepended dialogue increases, LLaVA shows a consistent performance drop for all three types of dialogues. Meanwhile, AIT successfully maintains the performance in General and Random types and shows less performance drop in Adversarial cases.

## 5.5 SANITY CHECK EXPERIMENTS: EFFECT OF PROMPT LENGTH TO HALLUCINATION

LLMs tend to degrade their performance as the prompt length increases (Liu et al., 2024). To check whether this happens in our experiments, we additionally conduct a sanity check experiment. In Table 7, we repeated each test question in EvalDial multiple times (N) to the LVLMs, and measured

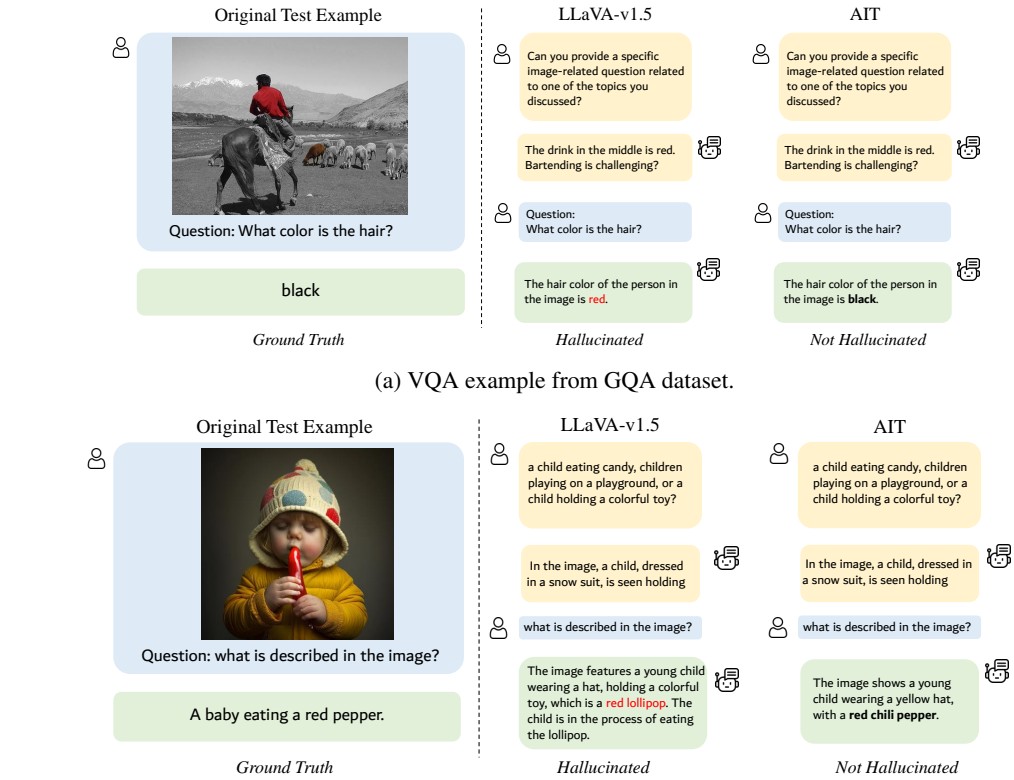

(a) VQA example from GQA dataset.

(b) Image Captioning example from Whoops dataset.

Figure 7: Visualization of generated examples by LLaVA and AIT. Hallucinated texts are in red.

the accuracy of the last question. Overall, prompt length (# repeats) has minimal impact on LVLM performance, indicating that content is more critical than length.

## 5.6 VISUALIZATION

Figure 7 visualizes two LVLM prediction examples with adversarial dialogues generated by AQG for VQA and Captioning task, respectively. Overall, while LLaVA-v1.5 hallucinates answers by preceding adversarial dialogues, AIT can generate correct answers without hallucinations. For example, as illustrated in Fig 7(a), as the preceding adversarial dialogue contains the word "red" (in yellow box), LLaVA unfaithfully answers "*the color of the person's hair*" as "*red*" in later chats (colored in red in the green box). Similarly, in Fig 7(b), the word "*candy*" in the adversarial preceding dialogue (in yellow box) hinders LLaVA from describing the image with the word "*lollipop*" in later chats (marked in red in the green box), which shows the LLaVA's weakness to the dialogue hallucination. On the other hand, although the same adversarial dialogue is prepended, AIT generates the correct answer or description of the image without hallucinations, by leveraging the power of robust fine-tuning against augmented adversarial dialogues. More visualization examples can be found in Appendix G.

## 6 CONCLUSION

In this work, we find that the existing instruction-following LVLMs are prone to be hallucinated by preceding user-system dialogues. To precisely validate this dialogue hallucination, we construct EvalDial, a large and diverse evaluation benchmark covering popular multi-modal datasets in VQA and captioning tasks, with a novel adversarial dialogue generator AQG. In addition, to mitigate such hallucination, we provide an in-depth analysis to help understand why such hallucination happens with input token attention analysis, and then propose AIT, a robust instruction-tuning method that maintains or even improves the zero-shot VQA and captioning performance of LVLMs in the presence of hallucinatory dialogues. We believe that our work can shed light on many applications requiring robust LVLMs such as the Red-teaming of visual-language assistants.

# 7 REPRODUCIBILITY STATEMENT

For reproducibility, we elaborate on the detailed generation process of our benchmark in Section 3.2, and its examples in Appendix A&B. The overall process of the proposed algorithms is explained in Section 4, and more detailed algorithm pseudocode is in Section C. Implementation details and hardware configurations are detailed in Section 5. We release our code at `https://anonymous.4open.science/r/LMM_hallucination-D52E/`.

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

(Supplementary Material)
# Mitigating Dialogue Hallucination for Large Vision Language Models via Adversarial Instruction Tuning

## A  GENERAL QUESTIONS GENERATED FROM GPT-3

In Table 8, we show 10 questions generated from GPT-3 using the prompt "Generate 10 general questions for a random image". All these general questions could be asked to any image.

Table 8: 10 general questions generated from GPT-3.

| Prompt: "Generate 10 general questions for a random image" |
|---|
| 1. "What is the geographical location depicted in the image" |
| 2. "Are there any identifiable landmarks or recognizable features in the image?" |
| 3. "What is the dominant color in the image?" |
| 4. "Are there any notable patterns or textures in the image?" |
| 5. "What is the source of light in the image (e.g., natural sunlight, artificial lighting)?" |
| 6. "Does the image evoke a sense of motion or stillness?" |
| 7. "What is the overall mood or atmosphere conveyed by the image?" |
| 8. "How does the image make you feel or what emotions does it elicit?" |
| 9. "What is the primary subject of the image?" |
| 10. What is the main point of focus or point of interest in the image?" |

## B  ADVERSARIAL QUESTIONS GENERATED BY AQG

In Figure 8 and Figure 9, we showed two examples of AQG-generated human-readable adversarial questions on VQA and Captioning task. Each question-related image and answer is also shown.

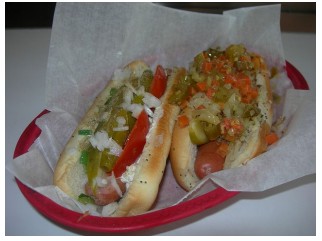

**Question**: Can you take a creative photo of a hot dog with various toppings, and capture some of the unique features of the condiments and toppings on the hot dog bun?

**Answer**: The hot dog is topped with a tomato, a pickle, and cheese.

Figure 8: Adversarial dialogue example for VQA task.

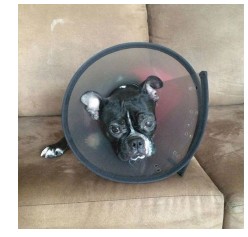

**Question**: What is the purpose of the cone around the dog's head?

**Answer**: In the image, a small dog is pictured laying on a couch with its head in a cone, possibly to prevent it from licking or scratching an irritated area or a recent surgical site on its head.

Figure 9: Adversarial dialogue example for Captioning task.

Table 9: Initial context prompt example of AQG.

| |
|---|
| Template: Generate an image-related question regarding |
| Context: small objects, background details, expected places, landmarks, related history, painting style, colors, and foods. |

Table 10: Additional results with FAITHSCORE on Captioning task. The higher, the better.

| Dialogue Type | LLaVA-v1.5 | | AIT | |
|---|---|---|---|---|
| | None | Adversarial | None | Adversarial |
| NoCaps | 0.89 | 0.75 | 0.89 | 0.87 |
| Flickr-30k | 0.91 | 0.80 | 0.91 | 0.90 |
| WHOOPS | 0.86 | 0.72 | 0.87 | 0.86 |

## C  DETAILED EXPLANATION OF THE OPTIMIZATION PROCEDURE OF AQG

**Context Prompt Initialization.** Table 9 shows the initialization of the context prompt we used for AQG. During optimization, only the context part is updated with the Gaussian random noise $\epsilon$ at the token level iteratively.

**Algorithm Explanation.** Algorithm 1 details the overall optimization process of AQG, which is self-explanatory. To find the best adversarial dialogue $\mathrm{X}^{adv}_{\mathtt{dialogue}}$ with higher target loss $\ell^{tgt}$, AQG starts with an initial prompt $\mathrm{X}^{init}_{\mathtt{prompt}}$ and proceeds the black-box optimization steps until round $r$ (in Lines 1–2). In each optimization step, it samples the gaussian noise $\epsilon$ and the noise injected tokens $\mathrm{Z}_{\mathtt{prompt}} = [\mathrm{Z}_{\mathtt{template}}; \mathrm{Z}_{\mathtt{context}} + \epsilon]$ is fed into the adversarial LVLM, generating the adversarial dialogue $\mathrm{X}^{adv}_{\mathtt{dialogue}}$ (in Lines 3–4). Next, the generated adversarial dialogue $\mathrm{X}^{adv}_{\mathtt{dialogue}}$ is fed into the original LVLM to hallucinate the answer $X^{adv}_{\mathrm{a}}$ (in Line 5). With the generated answer $\mathrm{X}^{adv}_{\mathrm{a}}$, we confirm to update the input tokens $\mathrm{Z}^{prompt} \leftarrow \mathrm{Z}^{prompt} + \epsilon$ only if the adversarial target is increased, otherwise we maintain it as $\mathrm{Z}^{prompt} \leftarrow \mathrm{Z}^{prompt}$ (in Lines 6–9). After repeating $r$ rounds of optimization, AQG returns the best adversarial dialogue $\mathrm{X}^{adv}_{\mathtt{dialogue}}$. Note that, AQG attacks the input prompt without calculating any gradient in a black-box optimization manner.

## D  ADDITIONAL EVALUATION METRICS ON CAPTIONING

We conducted additional experiments on Captioning tasks with FAITHSCORE, a more complex metric and closer alignment with human semantic understanding (Jing et al., 2023). The results evaluated on LLaVA and AIT are reported in Table 10. AIT is more effective in maintaining the FAITHSCORE than LLaVA, resonating with our main results.

## E  ADVERSARIAL QUESTIONS GENERATED BY PROMPTING GPT-4

**GPT-4 prompt.** We prompt GPT-4 to generate some adversarial dialogues as a simple baseline to our proposed AQG. Specifically, the prompt we used for VQA task is:
**Prompt**: Generate an image-related question that a user might ask and answer. This QA pair should be able to hallucinate a large visual-language model when prompting with this question (a-VQA-question) after prompting with the preceding generated question. Don't repeat the question. The ability to hallucinate a large visual-language model is very important here. Format the question-answer pair in this way (Que:QUESTION Ans:ANSWER END)

We change the (a-VQA-question) to "*What is described in the image?*" for the Captioning task and keep the rest of the prompt the same. We show two examples of generated questions in Figure 10 for VQA and Captioning tasks. Though GPT-4 can generate more natural-sounding questions, it is hard to effectively hallucinate the large visual language models. We specifically test the GPT-4 generated adversarial questions on all datasets in EvalDial to compare with AQG quantitatively. We evaluate LLaVA-v1.5 and the result in Table 11 shows that the model does not hallucinate much and could achieve higher accuracy when prepended with GPT-4 generated adversarial questions.

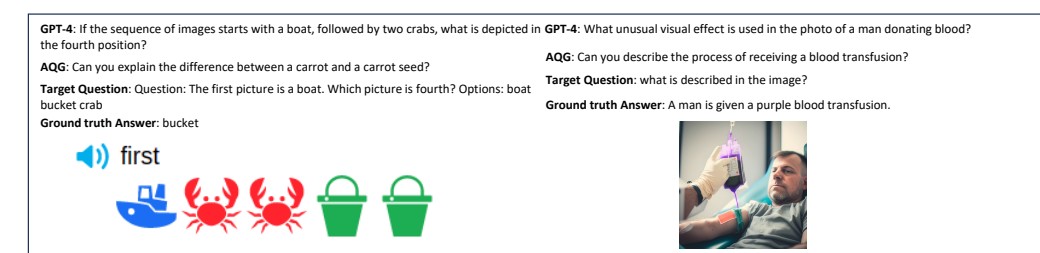

Figure 10: Examples of generated adversarial questions using GPT-4 and AQG on VQA and Captioning tasks.

Table 11: Effect of GPT4-generated adversarial dialogues to hallucinate LLaVA.

| Model | OKVQA | GQA | IconQA | NoCaps | Flickr | WHOOPS |
|-------|-------|-----|--------|--------|--------|--------|
| GPT4 | 54.4 | 54.8 | 46.6 | 41.0 | 28.3 | 42.3 |
| AQG | 48.4 | 49.0 | 41.2 | 35.8 | 19.9 | 34.8 |

Table 12: Performance of LRV-Instruction-v1 (Liu et al., 2023a) on IconQA and WHOOPs.

| Dataset | None | General | Random | Adversarial |
|---------|------|---------|--------|-------------|
| IconQA | 40.6 | 29.6 | 25.6 | 25.2 |
| WHOOPs | 33.1 | 36.4 | 26.8 | 16.2 |

## F    RESULT OF AN OBJECT HALLUCINATION BASELINE (LIU ET AL., 2023A) ON EVALDIAL

Because of the severe impact of hallucination on large visual language models, many mitigation methods have been proposed. We use (Liu et al., 2023a) as a baseline and evaluated on WHOOPs and IconQA datasets, and the result is shown in Table 12. We chose the LRV-Instruction v1 as it uses MiniGPT-4 as its backbone. Even though (Liu et al., 2023a) used a similar idea in fine-tuning as ours, the performance is not better or sometimes even worse than MiniGPT-4 due to different types of hallucinations. Their focus is on object hallucination, while our finding on dialogue hallucination could still confuse the large visual language model.

## G    MORE VISUALIZATIONS

Figure 11 shows additional visualization examples of dialogue hallucinations from NoCaps and OKVQA datasets, showing the effectiveness of our AIT compared to LLaVA. We highlighted the hallucinated part in red.

## H    EFFECT OF DATA SIZE FOR NON-ADVERSARIAL EXAMPLES USED DURING AIT

Since most training data do not contain adversarial dialogues, we explore the effect by directly including more non-adversarial prepended dialogue data from LLaVA-Instruct-665K during AIT. We add additional 50K, 100K, 150K, and 200K training data, and the result evaluated on IconQA is shown in Table 13. The performance increases with more non-adversarial examples but too much non-adversarial data also introduces noise, leading to performance fluctuation. With this ablation study, we believe including partial non-adversarial examples during the fine-tuning stage of AIT would help improve the performance.

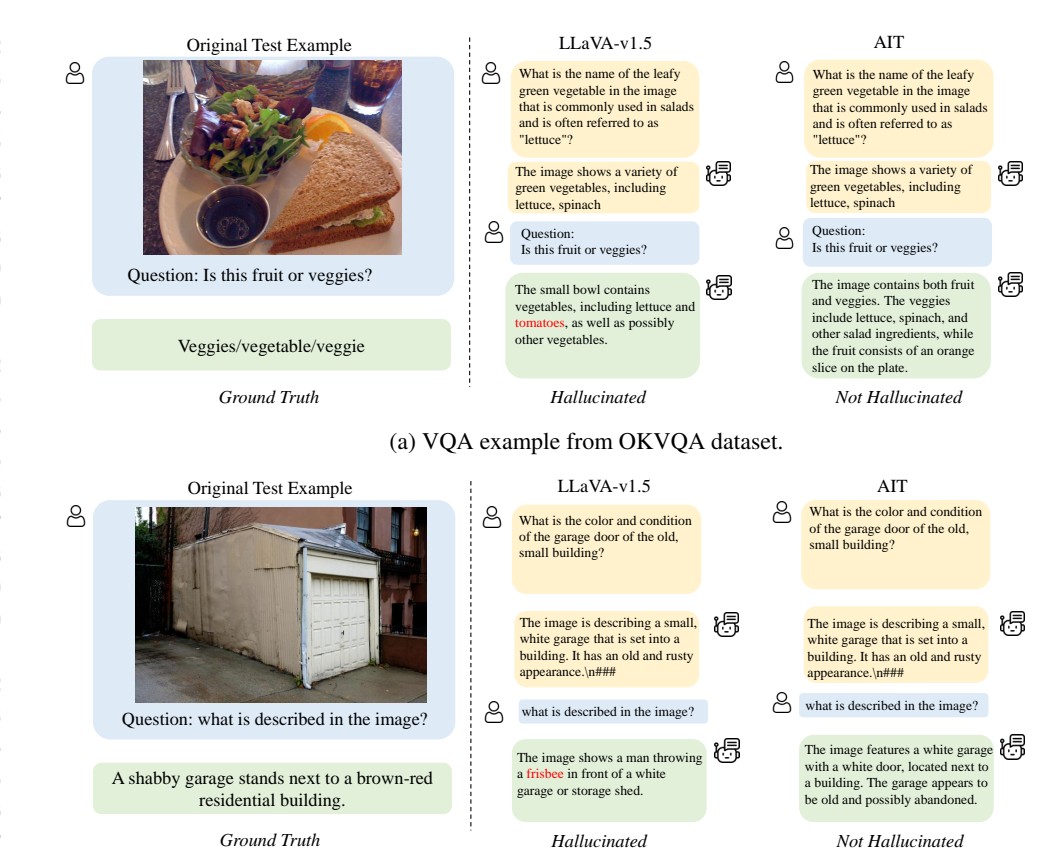

(a) VQA example from OKVQA dataset.

(b) Image Captioning example from NoCaps dataset.

Figure 11: Visualization of generated examples by LLaVA and AIT. Hallucinated texts are in red.

Table 13: Effect of adding non-adversarial data during AIT. Using a base AIT model, we include additional data from LLaVA-Instruct-665K that does not have adversarial prepended dialogues and evaluated on IconQA.

| Model | None | General | Random | Adversarial |
|-------|------|---------|--------|-------------|
| AIT | 45.8 | 34.4 | 44.6 | 41.4 |
| +50K | 47.2 | 47.0 | 46.8 | 45.2 |
| +100K | 45.8 | 47.4 | 46.6 | 45.6 |
| +150K | 47.2 | **49.0** | **48.0** | **47.8** |
| +200K | **49.8** | 44.6 | 48.0 | 46.4 |

