# OpenReview forum: "Mitigating Dialogue Hallucination for Large Vision Language Models via Adversarial Instruction Tuning"
_ICLR.cc/2025/Conference — Submitted to ICLR 2025_

### Official Review · Reviewer_k5BF · 2024-10-18

**Soundness:** 2
**Presentation:** 3
**Contribution:** 3
**Rating:** 5
**Confidence:** 3

**Summary:**

The paper introduces a new benchmark designed to evaluate vision-language models (VLM)  when dialogues are corrupted by adversarial questions. While this aims to address dialogue hallucination, such issues appear rare in real-world applications. Additionally, the benchmark is not tested on models known to hallucinate less, which raises concerns about its general applicability. There are also some contradictory statements about hallucinations and examples provided.

**Strengths:**

1. The paper presents a novel benchmark that specifically evaluates vision-language models when conversations are corrupted by adversarial questions from previous rounds. This benchmark is powered by the Adversarial Question Generator (AQG), which generates adversarial questions to test model robustness.

2. The paper proposes a solution, Adversarial Instruction Tuning (AIT), to mitigate the performance drop in VLMs caused by these adversarial dialogues, which is an important contribution to improving the robustness of these models.

**Weaknesses:**

1. The real-world applicability of adversarial questions, as demonstrated in the examples provided, seems limited. The likelihood of such adversarial scenarios occurring in practical use cases appears low, and hence, this might not be a widespread issue.

2. The hallucinated dialogues presented in Figure 2 may indicate that the underlying model itself is not robust enough. Many recent works, such as ShareGPT4V, have tackled hallucination issues with better training data. The paper does not evaluate its methods on models that are known to hallucinate less, leaving the effectiveness of the proposed method for stronger models unclear.

3. Section 3.1: Even by the paper’s own definition of hallucination (line 155: “the generated content that is nonsensical or unfaithful to the given source content”), the examples provided don’t clearly qualify as hallucinations. For instance, the responses in Figure 1 and Figure 2 make sense given the incorrect text from the previous dialogue. For example, in Figure 2 (mid-column), the answer "Antarctica" makes sense in relation to the mention of penguins from the prior round.

**Questions:**

None

---

> ### Author Response · Authors · 2024-11-22
>
> `W1. The real-world applicability of adversarial questions, as demonstrated in the examples provided, seems limited. The likelihood of such adversarial scenarios occurring in practical use cases appears low, and hence, this might not be a widespread issue.`
>
> Dialogue hallucinations represent a significant real-world issue, particularly in critical domains such as healthcare, legal, and financial services, where the user chats with the services throughout multiple round of conversations so accurate and trustworthy responses are paramount. For example, if a user with diabetes discusses having a transient cold and seeks advice on diabetes management, it would be highly dangerous if the treatment were hallucinated based on this context.
>
>
> `W2. Many recent works, such as ShareGPT4V, have tackled hallucination issues with better training data. The paper does not evaluate its methods on models that are known to hallucinate less, leaving the effectiveness of the proposed method for stronger models unclear.`
>
>
> Thank you very much for introducing the important relevant work. As suggested, we evaluted ShareGPT4V on EvalDial with GQA and Flickr-30k dataset and the results are shown below. **ShareGPT4V shows relatively better performance than LLaVA v1.5, but the performance on Adversarial dialogues are still not better than AIT**. This shows our proposed methods is still valuable in improving model's capability in mitigating hallucinations.
>
>
>
> | Model         | ShareGPT4V  |  |  |  | AIT  |  |  |  |
> |---------------|-----------------|--------------------|-------------------|------------------------|----------|-------------|------------|-----------------|
> | Dialogue Type | None            | General            | Random            | Adversarial            | None     | General     | Random     | Adversarial     |
> | GQA           | 58.6            | 57.4               | 58.0              | 55.4                   | 57.8     | 56.0        | 55.4       | 55.6            |
> | Flickr-30K    | 31.6            | 32.8               | 31.3              | 29.1                   | 39.5     | 38.8        | 38.2       | 38.7            |
>
>
> `W3. Section 3.1: Even by the paper’s own definition of hallucination (line 155: “the generated content that is nonsensical or unfaithful to the given source content”), the examples provided don’t clearly qualify as hallucinations. For instance, the responses in Figure 1 and Figure 2 make sense given the incorrect text from the previous dialogue. For example, in Figure 2 (mid-column), the answer "Antarctica" makes sense in relation to the mention of penguins from the prior round.`
>
> To be clear, line 155 “the generated content that is nonsensical or unfaithful to the given source content” is only a reference, but not part of our definition of the dialogue hallucination. We call a generated answer faithful **without** any dialogue but becomes unfaithful **after** some preceding dialogues, **dialogue hallucination**. In Figure 2, the original answer was "Asia". We acknowledge that there might be some ambiguity related to Figure 2. We presented a clearer example and updated Figure 2 to demonstrate three types of prepended dialogues.

---

> ### Author Response · Authors · 2024-11-25
>
> Dear Reviewer k5BF, we appreciate your valuable comments on our paper. We have prepared a rebuttal with an updated manuscript and tried our best to address your concerns. We notice that the author-reviewer discussion period is coming to an end, and we are willing to answer any unresolved or further questions you may have regarding our rebuttal if time is allowed.
> If our rebuttal has addressed your concerns, we would appreciate it if you would let us know your thoughts. Additionally, we will be happy to answer any further questions regarding the paper. Thank you for your time and consideration.

---

> > ### Author Response · Authors · 2024-11-28
> >
> > Dear Reviewer k5BF:
> >
> > Having not hear back from you, we are anxious to see whether there are any more concerns we can address. Please do let us know if there are anything further we can do to help resolve any questions.
> >
> > Thank you for your time.

---

### Official Review · Reviewer_RNog · 2024-10-30

**Soundness:** 3
**Presentation:** 3
**Contribution:** 2
**Rating:** 5
**Confidence:** 2

**Summary:**

This paper discovers that LVLMs can exhibit hallucination phenomena during user-system dialogues and presents a EvalDial benchmark for validation. Experimental results indicate that the performance of existing LVLMs significantly declines on this hallucination dataset. Furthermore, to mitigate the hallucination issue, the paper proposes an Adversarial Instruction Tuning method, which has been shown through experiments to reduce the occurrence of hallucinations effectively.

**Strengths:**

1. Benchmark on which the performance of existing LVLMs significantly declines.
2. Adversarial Instruction Tuning method, which has been shown through experiments to reduce the occurrence of hallucinations effectively.

**Weaknesses:**

Insufficient in-depth analysis of the experiment resulted in the inability to determine whether the dataset was meaningful. The benchmark data analysis is also insufficient. If the following issues in the QUESTIONS are solved, we will consider modifying the score.

1. Regarding the impact of prepended dialogues: Is the hallucination caused by an incorrect prediction made by the LVLM when the user asks a prepended question, which then leads to hallucinations in subsequent questions? Or is the hallucination triggered because the user's prepended question conflicts with the current question (i.e., unrelated to the answer)? Could you provide a detailed analysis of how the information in the preceding dialogues leads to hallucinations in the model?  It is essential to understand the mechanisms behind the mentioned hallucinations in LVLMs. Specifically, there are several following possible assumptions. (1)If the LVLM makes an incorrect prediction when interpreting a prepended question, this could indeed set off a chain reaction of hallucinations. The model might latch onto incorrect information, which it then carries forward into subsequent interactions. This could happen if the model's prediction is misled by the phrasing or context of the prepended question, causing it to deviate from the correct path of reasoning. Hallucinations occur when the model generates responses that are not grounded in the actual experiments. (2)Another hypothesis is that the model can not handle the long text of the prepended question. (3)A prepended question that is unrelated to the current question can also cause hallucinations. This is because the LVLM might struggle to recognize different segments of the dialogue. When the model encounters a question that seems to diverge significantly from the previous context, it may attempt to reconcile the two by creating a fabricated connection. This process can lead to the generation of wrong responses.
You may add specific experiments. For example, you could compare hallucination rates when prepended dialogues contain (1) incorrect information vs (2) unrelated information vs (3) correct, long, and related information.

2. In the benchmark, is a hallucinated prepended dialogue added before each question? If multiple rounds of dialogue are added before the current question, and the hallucinated dialogues are positioned differently within these rounds (e.g., at the beginning or just before the current question), or if the number of hallucinated dialogues varies, how would the model's performance change? You conduct ablation studies with different numbers of hallucinated dialogues and positions.

3. It is suggested to add a statistical table to analyze the benchmark, such as the distribution of dialogue types, and average dialogue length.

4. In Table 1, the mean DTAR values for hallucinated cases are 0.37 or 0.25, which are not particularly high. How does this prove that the hallucinations are caused by prepended dialogues? Furthermore, should the impact of the distance of the prepended dialogues from the current answer (whether closer or farther) also be considered in evaluating the influence on the answer? You may consider significance tests comparing hallucinated vs non-hallucinated cases. At the same time, it is suggested that the author test whether the DTAR value is affected by the relative position of the prepended hallucination dialogue and the current question.

**Questions:**

1. Regarding the impact of prepended dialogues: Is the hallucination caused by an incorrect prediction made by the LVLM when the user asks a prepended question, which then leads to hallucinations in subsequent questions? Or is the hallucination triggered because the user's prepended question conflicts with the current question (i.e., unrelated to the answer)? Could you provide a detailed analysis of how the information in the preceding dialogues leads to hallucinations in the model?

2. In the benchmark, is a hallucinated prepended dialogue added before each question? If multiple rounds of dialogue are added before the current question, and the hallucinated dialogues are positioned differently within these rounds (e.g., at the beginning or just before the current question), or if the number of hallucinated dialogues varies, how would the model's performance change?

3. It is suggested to add a statistical table to analysis the benchmark.

4. In Table 1, the mean DTAR values for hallucinated cases are 0.37 or 0.25, which are not particularly high. How does this prove that the hallucinations are caused by prepended dialogues? Furthermore, should the impact of the distance of the prepended dialogues from the current answer (whether closer or farther) also be considered in evaluating the influence on the answer?

---

> ### Author Response · Authors · 2024-11-22
>
> `W1&Q1. Could you provide a detailed analysis of how the information in the preceding dialogues leads to hallucinations in the model?`
>
> Thank you very much for your thorough assumptions and suggestions. From Table 2,  we observe that prepending the three types of dialogues degraded the model's ability to varying extents. We believe the hallucinations are caused by a **combination** of incorrect predictions of the current question and conflicts between the prepended questions and the current question. At your suggestion, we divide the Adversarial type dialogues into one of three types: 1) incorrect information, 2) unrelated information, and 3) long information using GPT-4o. The distribution table of these three types of dialogues for each OKVQA and NoCaps dataset is provided below.
>
> | Distribution |  |  |  |
> | - | - | - | - |
> | Dataset | Incorrect | Unrelated | Long |
> | OKVQA | 4.4% | 82.4% | 13.2% |
> | NoCaps | 9.4% | 59.2% | 31.4%|
>
> We then calculated the hallucination rates for the three types of preceding dialogues respectively on those datasets, and the results are shown in the Table below. From the table, LLaVA model shows similar performance across the three types of adversarial dialogues, with the worst results occurring with correct and long preceding dialogues. This is a valuable analysis and results, and we will include this part in our main paper to facilitate future study.
>
>
>
> | Model | LLaVA v1.5(7B) |  |  |
> | - | - | - | - |
> | Dataset | Incorrect | Unrelated | Long |
> | OKVQA | 47.5 | 49.3 | 45.0 |
> | NoCaps | 40.8 | 41.6 | 32.9|
>
>
> `W2&Q2. In the benchmark, is a hallucinated prepended dialogue added before each question? If multiple rounds of dialogue are added before the current question, and the hallucinated dialogues are positioned differently within these rounds (e.g., at the beginning or just before the current question), or if the number of hallucinated dialogues varies, how would the model's performance change?`
>
> Thank you for your constructive comments. In our benchmark, for a single-round prepended dialogue case, we prepended the hallucinated dialogue before the target question (See Table 2). For multi-rounds dialogue case, we added the hallucinated dialogues for every round of dialogue before the target question, which can measure the effect of varying number of hallucinatory dialogues (See Table 6). The results shows that **more prepended dialogues make the model less likely to answer the target question correctly**. In addition, as per your suggestion, we conducted additional experiment with different position of the adversarial dialogues and shows the result in Table below. For each tested QA pair, we prepended four rounds of dialogues with adversarial dialogues appear at different positions and use General type dialogues to fill in the rest of the dialogues. From the result, the **adversarial dialogue has more influence on the final accuracy as it positions closer to the final test question**. We will include this result in the final version, and again, thanks for helping us improve our manuscript.
>
>
> | Model | AIT | |  | |
> | - | - | - | - | - |
> | Adversarial Dialogue position | 1 | 2 | 3 | 4 |
> | Flickr-30K | 52.51 | 50.17 | 50.49 | 48.89 |
>
>
>
>
> `W3&Q3. It is suggested to add a statistical table to analyze the benchmark, such as the distribution of dialogue types, and average dialogue length.`
>
> Figure 6 of our draft illustrates the semantic distribution of each dialogue type within our benchmark. Specifically, it shows how far General, Random, and Adversarial dialogues are from the target question in the LVLM semantic space. The distribution indicates that adversarial questions are the closest to the target question, indicating them as subtle cases. Additionally, we provide the average token length of each dialogue type in our benchmark at the table below. We will add this statistics in the final version.
>
>
> | Model | General | Random | Adversarial |
> | ------------ | ---- | ---- | ---- |
> | Token Length | 13.7 | 25.1 | 27.8 |

---

> ### Author Response · Authors · 2024-11-22
>
> `W4&Q4. How does the DTAR values in Table 1 prove that the hallucinations are caused by prepended dialogues? It is suggested that the author test whether the DTAR value is affected by the relative position of the prepended hallucination dialogue and the current question.`
>
> We thank the reviewer for the careful comments. The DTAR score quantifies the proportion of influence of dialogue tokens over visual tokens. The DTAR usually **exhibits low values since LVLM is dominantly influenced by visual tokens** from the vision encoder. However, in the hallucinated case, the DTAR value increases so that it influences the latent features of the LVLM, hallucinating the output answer. That is, even with a seemingly **small increase** in absolute value, such as 0.19 to 0.37, it **can significantly change the output answer**, causing hallucination. For LLaVA, the high AUC-ROC score of 0.93 in Figure 4 indicates a substantial difference in DTAR score distribution between non-hallucinated and hallucinated cases.
>
> Furturemore, per your suggestion, we conducted additional experiments measuring the DTAR value on prepended adversarial dialogues at different positions in a multi-round setting. Specifically, we set the number of round to 2, and prepended an adversarial dialogue in the first or second position (with a General dialogue placed correspondingly in the second or first position) for LLaVA-v1.5 model. As shown in the table below, the DTAR scores of hallucinated cases are also higher than that of non-hallucinated case. Also, the closer the dialogue is to the target question, the greater its impact.
>
> | Cases            | Adv at 1st round | Adv at 2nd round |
> | ---------------- | --------------- | --------------- |
> | Non-hallucinated | 0.10 $\pm$ 0.04 | 0.13 $\pm$ 0.05 |
> | Hallucinated     | 0.17 $\pm$ 0.06 | 0.22 $\pm$ 0.07 |

---

> ### Author Response · Authors · 2024-11-25
>
> Dear Reviewer RNog, we appreciate your valuable comments on our paper. We have prepared a rebuttal with an updated manuscript and tried our best to address your concerns. We notice that the author-reviewer discussion period is coming to an end, and we are willing to answer any unresolved or further questions you may have regarding our rebuttal if time is allowed.
> If our rebuttal has addressed your concerns, we would appreciate it if you would let us know your thoughts. Additionally, we will be happy to answer any further questions regarding the paper. Thank you for your time and consideration.

---

> > ### Author Response · Authors · 2024-11-28
> >
> > Dear Reviewer RNog:
> >
> > Having not hear back from you, we are anxious to see whether there are any more concerns we can address. Please do let us know if there are anything further we can do to help resolve any questions.
> >
> > Thank you for your time.

---

### Official Review · Reviewer_k9D7 · 2024-11-02

**Soundness:** 3
**Presentation:** 3
**Contribution:** 2
**Rating:** 6
**Confidence:** 4

**Summary:**

This paper tackles the problem of hallucinations in Large Vision-Language Models (LVLMs), particularly how biases from prior dialogue contexts can lead to incorrect or "hallucinated" responses. To measure and address this, the authors introduce EvalDial, a benchmark created by adding adversarial dialogue sequences to existing multimodal datasets, using a novel Adversarial Question Generator (AQG). Their findings show that these adversarial dialogues significantly impact model accuracy. As a solution, they propose Adversarial Instruction Tuning (AIT), which fine-tunes LVLMs by masking dialogue tokens to reduce the bias from previous conversations. Experimental results demonstrate that AIT successfully minimizes hallucinations across multiple tasks while preserving overall model performance.

**Strengths:**

The introduction of EvalDial and AQG provides a structured and novel approach to testing model robustness against adversarial dialogue contexts, successfully inducing dialogue hallucinations across multiple models and confirming AQG's effectiveness. Notably, AIT demonstrates superior performance over other models in handling adversarial rounds, highlighting its potential as a reliable technique for mitigating dialogue-induced biases in broader LVLM applications where robustness is essential.

**Weaknesses:**

(1) The paper’s focus on hallucinated dialogue scenarios limits the ability to gauge the model’s general performance in non-hallucinated contexts. A broader evaluation that includes non-adversarial scenarios would provide a more comprehensive view of the model’s robustness and applicability.

(2) The study relies heavily on the CIDEr metric for evaluating image captioning, limiting insights into overall performance. Results using additional metrics such as BLEU, ROUGE, METEOR, or GPT-aided evaluation (accuracy) would provide a more balanced view of the model’s capabilities.

(3) While the Adversarial Dialogue is designed to be image-related and likely to induce hallucination in response to the original test question, it introduces content that is largely independent of the question itself. This raises the question: if the adversarial content is unrelated to the core test question, would simply resetting the chat between exchanges be a more straightforward solution?

**Questions:**

(1) The adversarial dialogue sequences used for prepending contain intentionally misleading answers, which likely disrupt baseline models due to their lack of exposure to such cases. Would AIT still demonstrate superior performance if prepend responses were correct, non-hallucinatory answers?

(2) Does the Masked Instruction Tuning approach risk overly discouraging reference to preceding dialogues? The relatively stable performance in Table 6, even as dialogue rounds increase, suggests this possibility. Could this approach prevent the model from effectively using relevant context from previous interactions, which is crucial in real dialogue scenarios where preceding exchanges often contain meaningful information?

(3) It would be valuable to understand how the model’s performance in captioning tasks changes as the length of dialogue rounds increases, particularly for AIT in handling longer, potentially more complex dialogue contexts.

---

> ### Author Response · Authors · 2024-11-22
>
> `W1. The paper’s focus on hallucinated dialogue scenarios limits the ability to gauge the model’s general performance in non-hallucinated contexts. A broader evaluation that includes non-adversarial scenarios would provide a more comprehensive view of the model’s robustness and applicability.`
>
> We thank the reviewer for this constructive comment. The model's overall performance in non-hallucinated contexts is already included in the "None" column of Table 2 in our draft. This refer to the non-adversarial scenarios without prepended dialogues, and our AIT shows comparable performance to the vanilla LLaVA-v1.5.
>
>
> `W2. The study relies heavily on the CIDEr metric for evaluating image captioning, limiting insights into overall performance. Results using additional metrics would provide a more balanced view of the model’s capabilities.`
>
> In Table 10 of Appendix D, our draft included the study with other captioning metrics, such as **FAITHSCORE**, which is a metric designed to evaluate the factual accuracy of outputs from LVLMs by verifying the consistency between generated text and visual inputs, and this metric demonstrates highly correlations with human judgments of faithfulness [1]. From the results, **AIT is still more effective** in maintaining the Faithscore than LLaVA v1.5, meaning that AIT's ability to reduce the dialogue hallucination is *consistent* with other metrics.
>
>
>
> ---
>
> [1] FaithScore: Fine-grained Evaluations of Hallucinations in Large Vision-Language Models, ArXiv, 2024.
>
>
> `W3. While the Adversarial Dialogue is designed to be image-related and likely to induce hallucination in response to the original test question, it introduces content that is largely independent of the question itself. This raises the question: if the adversarial content is unrelated to the core test question, would simply resetting the chat between exchanges be a more straightforward solution?`
>
> Thank you very much for your insightful comment. What you proposed could be a simple fix **if people are aware such dialogue hallucination exists**. However, there are still many scenarios where previous unrelated dialogue history is essential and valuable. For example, a patient previously discussed dietary habits and now reports a specific symptom. Understanding the patient's diet can help the LVLM identify potential causes of the symptom, leading to more accurate advice. It is often unclear whether a previous content is related or not to the core test question. Therefore, merely resetting the chat between exchanges is likely not an appropriate solution in many cases, and our method that guides robust responses against adversaries is necessary.
>
>
>
> `Q1.  Would AIT still demonstrate superior performance if prepend responses were correct, non-hallucinatory answers?`
>
> Our "General" type prepended dialogues could represent the correct, non-hallucinatory scenario, which is defined in Section 3.2. This type of question is about the general information of an image and each specific question is shown in Appendix A. As shown in Table 2 "Gen" column, **AIT still shows better performance than LLaVa v1.5 with General type dialogues prepended**.
>
>
>
> `Q2. Does the Masked Instruction Tuning approach risk overly discouraging reference to preceding dialogues?`
>
> This is an excellent question. We believe that our masked instruction tuning approach can actually **encourage models to identify appropriate references from preceding dialogues**. The adversarial question is unmasked and only the answer is masked, so our model is forced to learn the question-answer that is relevant to the target question. For example, in the training dataset, suppose there are preceding dialogues necessary to answer the target question, but there are also subtle adversarial dialogues prepended right before the target question. When we train an LVLM on this dataset, the model must identify the appropriate references in the preceding dialogues to answer the target question despite the presence of adversarial dialogues. This process may help the model become robust in distinguishing adversarial contexts and effectively finding relevant references.
>
>
> `Q3. It would be valuable to understand how the model’s performance in captioning tasks changes as the length of dialogue rounds increases.`
>
> Thanks for mentioning this experiment to make our paper more robust. We conducted multi-round experiments on Captioning task, particularly on Flickr-30k dataset and the result is shown in the Table below. Similar to the VQA task, more prepended adversarial dialogues make the model less likely to answer the target question correctly, and AIT maintains its performance on the Adversarial case. We will include this result in the final version.
>
> | Model | LLaVA v1.5(7B) |  |  | AIT  |  |  |
> | - | - | - | - | - | - | - |
> | Round | General | Random | Adversarial | General | Random | Adversarial |
> | 2 | 34.1 | 28.6 | 25.4 | 57.1 | 47.0 | 44.0 |
> | 4 | 33.9 | 27.5 | 22.6 | 56.3 | 43.8 | 45.2 |

---

> > ### Comment · Reviewer_k9D7 · 2024-11-23
> >
> > Thank you for your response; it has resolved many of my concerns.
> >
> > However, I noticed that most of the experiments in your work appear to be conducted using EvalDial (based on data used in LLaVA-Instruct-665k). At the same time, it seems that hallucinations were incorporated into LLaVA-Instruct-665k in a manner similar to the generation process of EvalDial, and the AIT model was trained on this data. While it is stated that the AIT model was trained with answers masked, there still seems to be a significant overlap between the training data for the AIT model and EvalDial. Could this be considered a fair comparison?

---

> > > ### Author Response · Authors · 2024-11-23
> > > **Official Response by Authors**
> > >
> > > Thanks for your prompt response; we are glad to hear that we have resolved many of your concerns.
> > >
> > > Regarding your follow-up question, while AIT employs AQG to introduce adversarial dialogue hallucinations into the training dataset similar to EvalDial, the training set for LLaVA-Instruct-665k and the test set for EvalDial (comprising OKVQA, GQA, IconQA, NoCaps, Flickr-30K, and WHOOPS) consist of **different** image-text pair examples; there is **no overlap** between examples in the AIT model's training set and EvalDial's test set. Therefore, our experiment demonstrates that the dialogue hallucination learned from the training set **generalizes to unseen** examples in the test set. This is consistent with the typical evaluation process in machine learning and thus constitutes a fair comparison. We hope this answers your question and address all your concerns.

---

### Official Review · Reviewer_2MCt · 2024-11-03

**Soundness:** 3
**Presentation:** 4
**Contribution:** 3
**Rating:** 6
**Confidence:** 3

**Summary:**

This paper addresses the issue of dialogue hallucination in LVLMs, where preceding dialogues can lead to unreliable outputs.

It introduces a benchmark dataset, EvalDial, using the Adversarial Question Generator from VQA and image captioning datasets. Then, it proposes the Adversarial Instruction Tuning method, which fine-tunes LVLMs against hallucinatory dialogues.

The results suggest most of the well-performing LVLM failed on the EvalDial benchmark, in which AIT successfully reduced dialogue hallucination while maintaining performance.

Overall, this work aims at an important research problem, presents a reasonable and effective approach, and contains insightful discussions. The implementation is reproducible.

**Strengths:**

- The AQG is a reasonable and scalable algorithm for data generation.
- The experiment design is comprehensive, and the results show the AIT approach effectively mitigates dialogue hallucination.
- There is a potential that both AQG algorithm and AIT strategy can be extended to more tasks and modalities.

**Weaknesses:**

An important contribution of the work is to propose AIT as a method to mitigate dialogue hallucination. However, the evaluation only focuses on the EvalDial benchmark, which includes synthetic dialogue augmented from non-dialogue data. This is insufficient to justify AIT's real-world value in countering the visual dialogue hallucination problem.

**Questions:**

Does AQG-generated data somehow reflect the real distribution of LVLM's hallucination behaviors? Or is it just a way to create fake adversarial dialogue that serves AIT's finetuning purpose?

---

> ### Author Response · Authors · 2024-11-22
>
> We sincerely appreciate the reviewers' constructive comments and positive feedback on our manuscript.
>
> `Q1. Does AQG-generated data somehow reflect the real distribution of LVLM's hallucination behaviors? Or is it just a way to create fake adversarial dialogue that serves AIT's finetuning purpose?`
>
> As our objective listed in Section 3.3, AQG works to generate natural sounding dialogs. The distribution of real-world dialog are vast. However, we expect them to be natural sounding, and therefore AQG should be effective even in real world use cases.

---

> > ### Comment · Reviewer_2MCt · 2024-11-25
> > **Review update**
> >
> > After carefully reading other reviewers' comments (especially from reviewer k5BF) and author's responses, my concerns about the issues mentioned in weakness have intensified. Therefore, I lowered my recommendation to the borderline; but overall, I'm still slightly on the positive side.

---

> > > ### Author Response · Authors · 2024-11-25
> > >
> > > We regret to hear that your concerns have intensified. Our experiment on real-world cases is presented in the "None" column of Table 2. Specifically, each test question *does not* have preceding dialogues, meaning they do not include any extra synthetic dialogue information but serve as a real-world scenario. From the results, AIT shows better performance than the original LLaVA v1.5 model across most datasets. We hope you may reconsider your decision, and we would be grateful for the opportunity to address any further questions or concerns you may have.

---

> > > > ### Author Response · Authors · 2024-11-28
> > > >
> > > > Dear reviewer 2MCt:
> > > >
> > > > We hope that our last explanation regarding the 'None' column reflecting real world data, and that AIT shows better performance on this real world data has resolved your concern on AIT's real world value. We are anxious to hear back on this and whether there is anything more we can do to address your concern here.

---

> > > > > ### Author Response · Authors · 2024-12-04
> > > > > **Further clarification**
> > > > >
> > > > > Dear Reviewer 2MCt,
> > > > >
> > > > > To further clarify your concerns, we conducted an additional experiment to evaluate how natural the AQG-generated adversarial dialogues are compared to real distributions of multi-turn conversations. The results show that **our AQG-generated dialogues are as natural as the real conversations**. Below are the details of our experiments:
> > > > >
> > > > > **Setup.** We are based on the LLAVA-Instruct-80k dataset, consisting of natural multi-turn dialogues generated by GPT that closely mimic a real-world conversation given an image.
> > > > > Firstly, we identified a subset S1 from the LLAVA-Instruct-80k containing samples with multi-turn preceding dialogues where LLaVA-v1.5 hallucinates on the last question but does not hallucinate without preceding dialogues. Then, for each sample in S1, we generated preceding adversarial dialogues using AQG, and further extracted another subset S2 that LLaVA hallucinates with both the original multi-turn dialogues and the AQG-generated adversarial dialogues.
> > > > >
> > > > > On the extracted hallucinatory subset S2, we have **two sets of preceding dialogues, one from the LLAVA-Instruct-80k and another generated by AQG**. Finally, we compare the naturalness of these preceding dialogues. Specifically, we asked GPT-4o to assess the **naturalness of each set of multi-turn preceding dialogues** by using the detailed instruction with a scoring rubric as below:
> > > > >
> > > > > LLM instruction (for measuring the naturalness of dialogues):
> > > > > > You are an LLM judge to measure the naturalness of a multi-turn dialogue. Your task is to determine how natural the given multi-round QA conversation is on a scale of 5.
> > > > > > - 5: this dialogue is very natural and consists of consistent Q&A about an image that a human generate
> > > > > > - 4: this dialogue is natural and could be loosely about the image
> > > > > > - 3: each round of conversation could still be generated by humans even if the content might not be related to each other round
> > > > > > - 2: this dialogue is not natural and can hardly be expressed by a general human being
> > > > > > - 1: this dialogue is very unnatural and resembles an adversarial attack intended to break the system rather than something a human would do.
> > > > > >
> > > > > > Answer with only a digit from 1 to 5.
> > > > >
> > > > >
> > > > > **Result.** The result is shown in the table below. The original LLaVA dataset is rated with 4.46 while AQG generated dialogues is rated with 4.22, indicating that the AQG generated dialogues are highly natural as the original LLaVA conversations. Overall, AQG-generated data is still shows acceptible ability in generating natural-sounding multi-turn conversations.
> > > > >
> > > > > | Model | LLAVA | AQG |
> > > > > | ------------ | ---- | ---- |
> > > > > | Naturalness | 4.46 | 4.22 |
> > > > >
> > > > > We hope this clarifies your original concerns, and that we can regain your positive support for our manuscript.
> > > > >
> > > > > Authors

---

### Author Response · Authors · 2024-11-22

We sincerely appreciate the reviewers' positive feedback and valuable comments. Most reviewers agreed that (1) **the methodology is novel and effective** (All reviewers), (2) **the empirical results are strong** (Reviewer 2MCt, k9D7, and RNog), (3) **the proposed benchmark is novel and comprehensive** (Reviewer 2MCt, RNog, and k5BF), and (4) **the proposed approach is scalable and broadly applicable** (Reviewer 2MCt and k9D7). During the rebuttal, we addressed the reviewers' remaining concerns by providing clarifications with additional experimental results (including one latest LVLM backbone and more root-cause studies). We hope that the remaining concerns are successfully addressed by the rebuttal and are happy to answer more questions during the discussion period.

---

### Meta-Review · Area_Chair_vrHa · 2024-12-21

**Metareview:**

### Summary:
This paper proposes a method to mitigate dialogue hallucination in Large Vision Language Models (LVLMs) through:
1. A benchmark called EvalDial that extends existing datasets with adversarial dialogues
2. An Adversarial Instruction Tuning (AIT) approach to make LVLMs more robust against hallucination
3. Experimental results showing AIT can reduce hallucination while maintaining performance

### Strengths:
- Novel benchmark creation:
> "The paper presents a novel benchmark that specifically evaluates vision-language models when conversations are corrupted by adversarial questions from previous rounds. This benchmark is powered by the Adversarial Question Generator (AQG), which generates adversarial questions to test model robustness."

- Initial empirical validation:
> "The experiment design is comprehensive, and the results show the AIT approach effectively mitigates dialogue hallucination."

### Weaknesses:
- Limited real-world applicability:
> "The real-world applicability of adversarial questions, as demonstrated in the examples provided, seems limited. The likelihood of such adversarial scenarios occurring in practical use cases appears low, and hence, this might not be a widespread issue."

- Insufficient evaluation scope:
> "The paper's focus on hallucinated dialogue scenarios limits the ability to gauge the model's general performance in non-hallucinated contexts. A broader evaluation that includes non-adversarial scenarios would provide a more comprehensive view of the model's robustness and applicability."

- Methodological concerns:
> "Many recent works, such as ShareGPT4V, have tackled hallucination issues with better training data. The paper does not evaluate its methods on models that are known to hallucinate less, leaving the effectiveness of the proposed method for stronger models unclear."


Justification: While the paper presents an interesting direction for addressing dialogue hallucination in LVLMs, the weaknesses in real-world applicability, evaluation methodology, and comparison with state-of-the-art solutions outweigh its contributions. The authors' responses during discussion, while thorough, did not sufficiently address these fundamental concerns. I recommend rejection with encouragement to address these issues for future submission.

**Additional Comments On Reviewer Discussion:**

During the rebuttal period, the authors made efforts to address reviewer concerns by:
1. Providing additional experiments on real-world cases and model comparison with ShareGPT4V
2. Adding statistical analysis of dialogue types and their impact
3. Clarifying the training methodology and dataset separation

However, the core concerns about real-world applicability and effectiveness compared to simpler solutions remained unresolved. While reviewer 2MCt lowered their score after discussion, reviewers k9D7 and RNog's major concerns about methodology and evaluation scope were not fully addressed.

---

### Decision · Program_Chairs · 2025-01-22

Reject